# Engineering the green algae *Chlamydomonas incerta* for recombinant protein production

**Kalisa Kang**[1], **Évellin do Espirito Santo**[1,2], **Crisandra Jade Diaz**[1], **Aaron Oliver**[3], **Lisa Saxton**[1], **Lauren May**[4], **Stephen Mayfield**[1,5], **João Vitor Dutra Molino**[1]*

1 Department of Molecular Biology, School of Biological Sciences, University of California San Diego, La Jolla, California, United States of America, 2 Department of Biochemical and Pharmaceutical Technology, Faculty of Pharmaceutical Sciences, University of São Paulo, São Paulo, Brazil, 3 Center for Marine Biotechnology and Biomedicine, Scripps Institution of Oceanography, University of California San Diego, La Jolla, California, United States of America, 4 Biological Sciences Department, California Polytechnic State University, San Luis Obispo, California, United States of America, 5 Algenesis Materials, San Diego, California, United States of America

* jdutramolino@ucsd.edu

## Abstract

*Chlamydomonas incerta*, a genetically close relative of the model green alga *Chlamydomonas reinhardtii*, shows significant potential as a host for recombinant protein expression. Because of the close genetic relationship between *C. incerta* and *C. reinhardtii*, this species offers an additional reference point for advancing our understanding of photosynthetic organisms, and also provides a potential new candidate for biotechnological applications. This study investigates *C. incerta*'s capacity to express three recombinant proteins: the fluorescent protein mCherry, the hemicellulose-degrading enzyme xylanase, and the plastic-degrading enzyme PHL7. We have also examined the capacity to target protein expression to various cellular compartments in this alga, including the cytosol, secretory pathway, cytoplasmic membrane, and cell wall. When compared directly with *C. reinhardtii*, *C. incerta* exhibited a distinct but notable capacity for recombinant protein production. Cellular transformation with a vector encoding mCherry revealed that *C. incerta* produced approximately 3.5 times higher fluorescence levels and a 3.7-fold increase in immunoblot intensity compared to *C. reinhardtii*. For xylanase expression and secretion, both *C. incerta* and *C. reinhardtii* showed similar secretion capacities and enzymatic activities, with comparable xylan degradation rates, highlighting the industrial applicability of xylanase expression in microalgae. Finally, *C. incerta* showed comparable PHL7 activity levels to *C. reinhardtii*, as demonstrated by the in vitro degradation of a polyester polyurethane suspension, Impranil® DLN. Finally, we also explored the potential of cellular fusion for the generation of genetic hybrids between *C. incerta* and *C. reinhardtii* as a means to enhance phenotypic diversity and augment genetic variation. We were able to generate genetic fusion that could exchange both the recombinant protein genes, as well as associated selectable marker genes into recombinant offspring. These findings emphasize *C. incerta*'s potential as a robust platform for recombinant protein production, and as a powerful tool for gaining a better understanding of microalgal biology.

**Data availability statement:** All data, figure, and supplementary materials are available at Zenodo. DOI: 10.5281/zenodo.13948022 (https://doi.org/10.5281/zenodo.13948021)

**Funding:** This material is based upon work supported by the U.S. Department of Energy's Office of Energy Efficiency and Renewable Energy (EERE) under the APEX award number DE-EE0009671. This material was also supported by the S10OD030505 and NINDS P30NS047101 grants from the UCSD microscopy core. There was no additional external funding received for this study. The funder provided support in the form of salaries for authors but did not have any additional role in the study design, data collection and analysis, decision to publish, or preparation of the manuscript. The specific roles of authors are articulated in the 'Author Contributions' section.

**Competing interests:** SM was a founding of and holds an equity stake in Algenesis Inc, a company that could potentially benefit from this work. The remaining authors declare that the research was conducted in the absence of any commercial or financial relationships that could be construed as a potential conflict of interest. This does not alter our adherence to PLOS ONE policies on sharing data and materials.

## Introduction

The development of production processes for recombinant proteins was a pivotal advancement in biotechnology. Recombinant proteins have become the dominant form of complex therapeutics, called biologics, and are essential tools for studying biological processes and advancing our understanding of cells and organisms. *Escherichia coli*, *Saccharomyces cerevisiae*, *Komagataella pastoris*, Chinese hamster ovary (CHO) cells, and human embryonic kidney (HEK293) cells, are some of the most developed recombinant protein production systems currently used. *E. coli* was the first, and remains the most commonly used bacterial host for expression of simple recombinant proteins [1]. Cytoplasmic protein expression in *E. coli* can reach up to 50% of the total cellular protein [2]. However, the inability of *E. coli* to efficiently secrete recombinant proteins into the extracellular medium and a general lack of post-translational modifications, such as N-linked and O-linked glycosylation, amidation, hydroxylation, myristoylation, palmitoylation, and sulfation, remain major drawbacks for industrial production of many proteins in bacteria. Yeast systems, such as *S. cerevisiae* and *K. pastoris*, provide similar protein production to *E. coli* with the additional benefits of a eukaryotic system: improved protein folding and addition of most post-translational modifications [3,4]. *K. pastoris* has become more widely used due to higher levels of recombinant protein expression compared to *S. cerevisiae* [5]. However, as a methylotrophic yeast, large-scale cultivation of *K. pastoris* poses physical safety hazards due to a low percentage of methanol in induction media, which can be flammable [6]. Mammalian expression systems are primarily used to generate secreted recombinant proteins. They produce complex N-linked and O-linked glycan structures on recombinant proteins that are not only dependent on the protein but also on the mammalian cell type used. Serum-free media have been developed from CHO and HEK293 cell lines, simplifying the purification of secreted recombinant proteins. However, the high cost of the media for these cell lines makes large-scale bioproduction costly, especially since mammalian cell cultures are susceptible to viral contamination [7,8].

Microalgae represent another promising host for recombinant protein production, offering unique advantages over the aforementioned traditional systems. The genetic engineering of microalgae holds transformative potential for a range of biotechnological applications, including the synthesis of recombinant proteins, biofuels, and bioplastics [9–12]. However, to fully harness this potential, the current genetic tools and methodologies for these organisms need substantial enhancement. Microalgae are already vital to industries such as animal and fish feed, cosmetics, pigments, and nutraceuticals [13–17]. *Chlamydxomonas reinhardtii* prevails as a model organism due to its well-established classical genetics and more recent molecular technologies, such as transformation for both nuclear and chloroplast genomes [18–21].

Nuclear transgene expression is advantageous over chloroplast expression primarily because it not only allows for post-translational modifications, but also allows for targeting protein expression to various cellular compartments, including the cytosol, secretory pathway, cytoplasmic membrane, and cell wall, with the appropriate secretion and localization tags. Despite these benefits, the molecular mechanisms responsible for the poor nuclear transgene expression in *C. reinhardtii* remain insufficiently understood. Challenges such as suboptimal promoters, positional effects from random genome integration, and transgene silencing significantly impede progress [22,23]. Nuclear transformation in *C. reinhardtii* typically occurs via random insertion through non-homologous end joining, resulting in variable expression levels that are influenced by the surrounding genomic regions [24]. Moreover, transgene silencing at both transcriptional and post-transcriptional levels poses a major hurdle [22]. Efforts to enhance nuclear transgene expression in *C. reinhardtii* include UV mutagenesis, leading to strains like UVM4 with improved transgene expression [25]. Yet, recombinant protein accumulation achieved in this strain remains low by industrial standards, with reported

levels reaching only 10 mg/L [26], which is considered the minimum concentration for any commercial process [27,28]. This highlights the pressing need for innovative strategies to achieve higher and more consistent transgene expression.

In contrast, *Chlamydomonas incerta* has not yet been developed as a platform for recombinant protein expression, despite being the genetically closest species to *C. reinhardtii* [29]. The absence of established mating types, transformation methods, and regulatory elements indicates a need for genetic engineering efforts in *C. incerta*. Here, we transformed *C. incerta* with nuclear expression secretion vectors, originally developed for *C. reinhardtii* (pJP32mCherry, pJP32Xylanase, pJP32PHL7), that contain various recombinant proteins (i.e., mCherry, xylanase 1, and a plastic-degrading enzyme [PHL7], respectively) fused to the bleomycin antibiotic resistance gene via the foot-and-mouth-disease-virus (FDMV) 2A peptide (F2A) and signal peptide (SP7) sequences [30–32]. We also transformed *C. incerta* with cytosolic, cell membrane, and cell wall mCherry expression vectors (pAH04mCherry, pJPM1mCherry, pJPW2mCherry, respectively). We identified that *C. incerta* expressed, localized, and secreted these recombinant proteins at intrinsically high levels, demonstrating that *C. incerta* has the potential to be a robust platform for biotechnological applications. We also demonstrate that *C. incerta* can undergo interspecific hybridization with *C. reinhardtii*, allowing us to increase genetic diversity with the potential to acquire new desirable traits, such as enhanced environmental tolerance. The development of recombinant strains of *C. incerta* represents a significant opportunity to propel algal research and biotechnology forward. This approach addresses current limitations and opens new pathways for sustainable and economically viable microalgal production processes, ultimately driving innovation and efficiency across various industries.

## Results

### Targeting of mCherry to the cytosol, secretory pathway, cell membrane, and cell wall in *C. incerta*

To explore the potential for expressing and targeting recombinant proteins in *C. incerta*, vectors originally designed for *C. reinhardtii* were utilized to express three recombinant proteins in *C. incerta*. mCherry was expressed in the cytosol via the pAH04mCherry vector, in the cell membrane via the pJPM1mCherry vector, and in the cell wall via the pJPW2mCherry vector. The pAH04mCherry vector was based on the direct fusion of the bleomycin resistance gene and F2A self-cleaving peptide sequence with mCherry protein gene, as previously described [30]. The ribosomal skipping caused by the 2A peptide results in separation of both protein moieties, releasing mCherry in the cytosol. The pJPM1mCherry vector was assembled with the pJP22 backbone [30] and contains the SP2 ARS1 signal peptide from *C. reinhardtii*, targeting the protein to the endoplasmic reticulum for processing, along with MAW8 containing a putative GPI anchoring site for lipidation and entrapment to the membrane [33,34]. For cell wall expression, the pJPW2mCherry vector was created using the pJP32 backbone, with the signal peptide SP7, a glycine-serine flexible linker and hydrophilic linker to connect mCherry to GP1, causing mCherry to be anchored to the cell wall alongside GP1, a major protein component of *C. reinhardtii* cell wall [34,35].

For each vector, transformant colonies were picked into 96-well plates, grown for 7 days, and fluorescence readings were measured to screen for positive transformants. The top performing fluorescent strains were selected for further analyses. Fluorescent microscopy revealed distinct localization patterns of the mCherry protein compared to the wild type cells (Fig 1A). In cells with cytosolic mCherry expression, fluorescence was observed, outlining the nucleus and possibly two contractile vacuoles (Fig 1B). Additionally, there was a clear

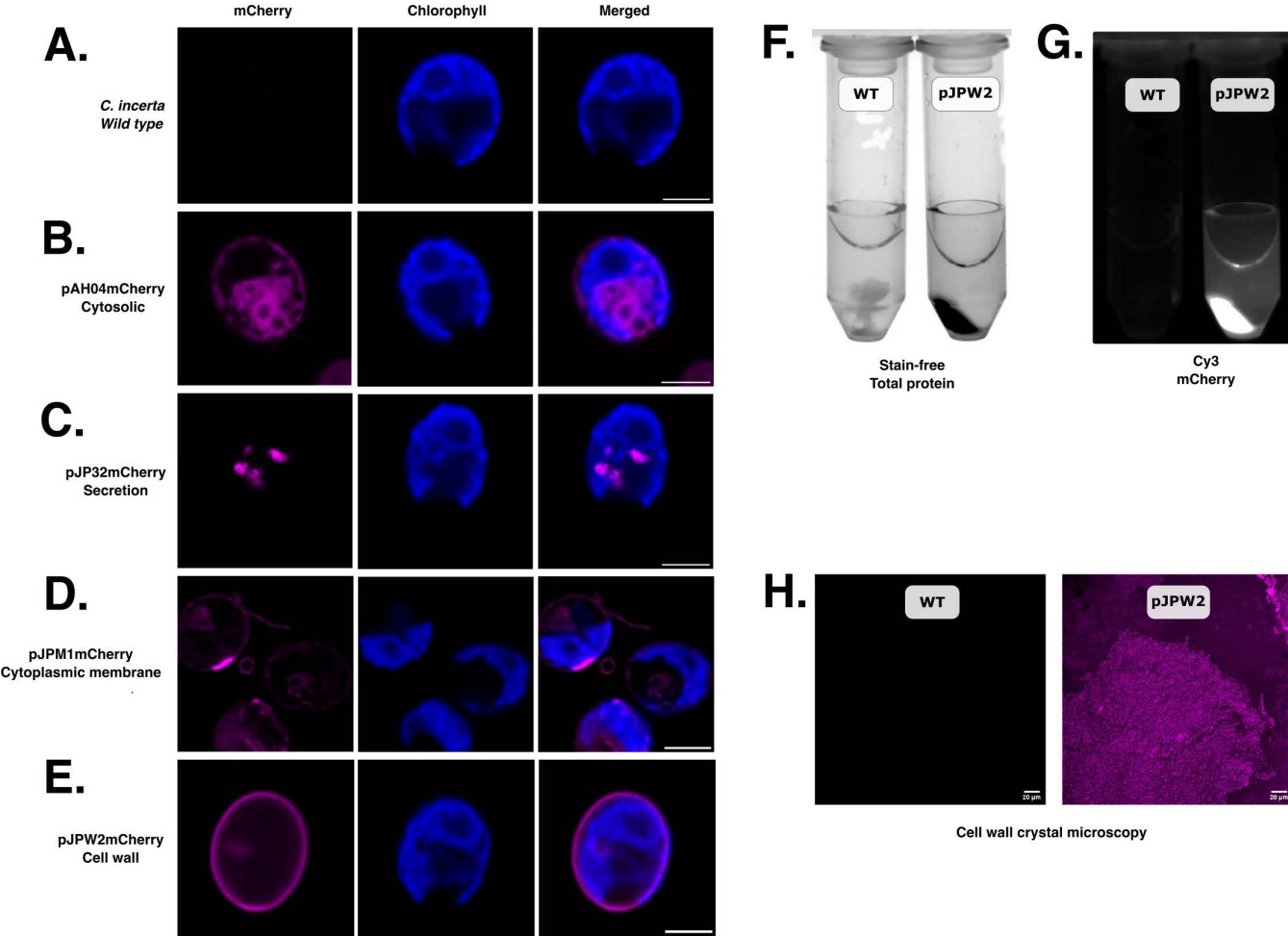

**Fig 1. Cellular localization of mCherry to the cytosol, cell membrane, and cell wall.** Fluorescent microscopy demonstrated mCherry expression in the B) cytosol, C) secretory pathway, D) cell membrane, and E) cell wall. Scale bars indicate 5 μm. F, G, H) The recrystallized cell wall components of transgenic *C. incerta* pJPW2mCherry showed mCherry fluorescence, absent on the wild type. The recombinant strains were deposited at Chlamydomonas collection (Ci_pAH04mCherry CC-6206; Ci_pJP32mCherry CC-6207; Ci_pJPM1 CC-6208; Ci_pJPW2 CC-6209).

separation between the chloroplast and cytosolic mCherry, which is observed by the different localization of chlorophyll and mCherry fluorescence. With the secretion vector, mCherry was observed inside secretory vesicles, as expected for an mCherry-secreting strain (Fig 1C). Targeting mCherry to the cell membrane resulted in fluorescence being evident not only in the cytoplasmic membrane, but also around the cell flagella [36] (Fig 1D). To further investigate mCherry localization within the cell membrane, a video was recorded of the mCherry-expressing cells in motion, capturing fluorescence in the membrane and flagella as the cells swam (Video 1). In the cell wall-targeted expression, mCherry fluorescence was successfully visualized in the expected position of the cell wall (Fig 1E). Following this, the cell wall proteins were extracted using perchlorate, recrystallized using diafiltration, and visualized using UV and LED light sources and fluorescent microscopy as described in the Methods section (Figs 1F–1H). Overall, the expression and visualization of mCherry in the cytosol, cell membrane, and cell wall demonstrated the versatility and robustness of the recombinant

protein expression in *C. incerta* and *C. reinhardtii*. The successful targeting and clear fluorescence signals in each compartment underscored the potential of these algae as platforms for recombinant protein production and as tools to explore microalgae biology.

## Improved secretion and expression of mCherry fluorescent protein in *C. incerta* compared to *C. reinhardtii*

To determine the potential of *C. incerta* as a foundation for secretion of recombinant proteins, three proteins with a variety of applications were chosen: mCherry, xylanase, and PHL7. Starting with mCherry, *C. incerta* and *C. reinhardtii* were transformed with the pJP32mCherry vector in triplicates. For *C. incerta* and *C. reinhardtii*, an average of 276 and 1848 colonies, respectively, were obtained for each transformation (S1 Fig). When normalizing with the cell density using chlorophyll fluorescence, mCherry fluorescence for *C. incerta* was approximately 3.5 times higher than that of *C. reinhardtii* (adjusted p-value < 2.2 x 10^{-16}), and 3.9 times higher by overall mCherry signal (adjusted p-value < 2.2 x 10^{-16}). (Fig 2A). *C. incerta*

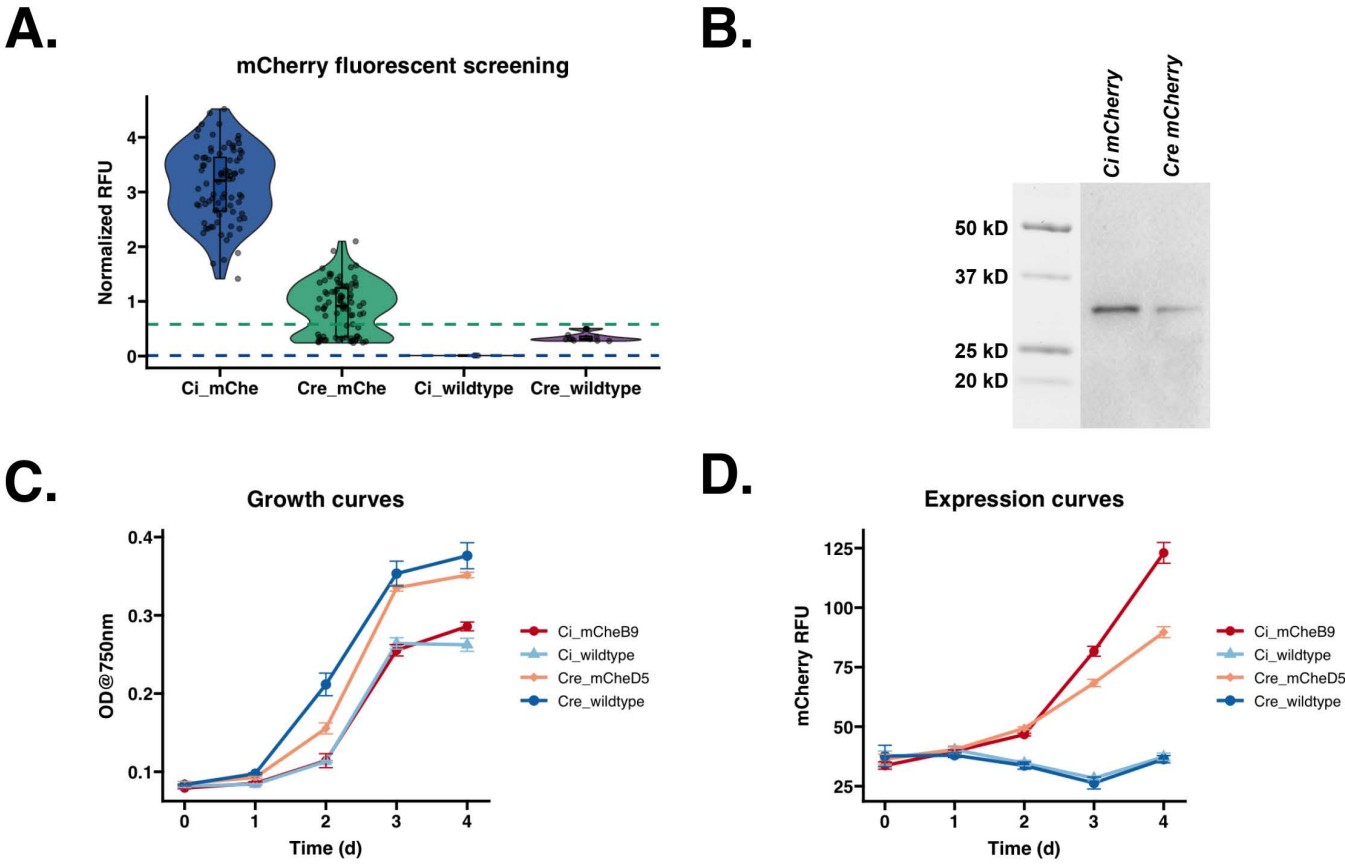

**Fig 2.** ***C. incerta*** exhibits higher secretion and expression of mCherry compared to ***C. reinhardtii***. **A)** mCherry fluorescence readings at excitation wavelength of 580/9 nm and emission wavelength of 610/20 nm for ***C. incerta*** (n = 84) and ***C. reinhardtii*** (n = 84) demonstrated a 3.5-fold higher fluorescence intensity for ***C. incerta*** than ***C reinhardtii*** (p-value < 2.2 x 10^{-16}). Fluorescence readings were normalized with cell density using chlorophyll fluorescence. Threshold for positive expression (blue dashed line at 9.10 x 10^{-3} RFU for ***C. incerta*** and green dashed line at 0.580 RFU for ***C. reinhardtii***) was defined as three standard deviations above the mean activity level observed in the wild types. **B)** Immunoblot analysis using an anti-RFP antibody indicated the expected band size of mCherry at approximately 30 kDa for both ***C. incerta*** and ***C. reinhardtii***, with a 3.76-fold higher band intensity for ***C. incerta***. **C)** Fluorescent microscopy images showed mCherry signals inside vesicles localized within the cell's secretory pathway. **D, E)** Growth curves of the top-expressing clone indicated that mCherry secretion from the transgenic ***C. incerta*** line did not affect cell growth compared to transgenic ***C. reinhardtii*** lines and their respective wild types.

demonstrated a higher percentage of positive transformants expressing mCherry, achieving 100% (84/84), compared to *C. reinhardtii*, which exhibited a lower rate of 67.470% (56/83) (Fig 2A). The threshold for positive expression was defined as three standard deviations above the mean fluorescence observed in the wild type strains. The three highest-expressing clones from both strains were selected for further analyses. Immunoblot analysis using an anti-RFP antibody indicated the expected band size of mCherry (with post-translational modifications) at approximately 30 kDa for both *C. incerta* and *C. reinhardtii*, with a 3.8-fold higher band intensity for *C. incerta*, in agreement with the results from the fluorescence measurements (Fig 2B). Growth curves of the top-expressing clones indicated that mCherry secretion from the transgenic *C. incerta* and *C. reinhardtii* lines did not affect cell growth compared to their respective wild type parent. Despite the fact that both the transgenic and wild type *C. reinhardtii* strains achieving higher cell densities then *C. incerta*, mCherry production was 290% higher for *C. incerta* strains than in the *C. reinhardtii* strains, when assessed by direct mCherry fluorescence, or 250% when accounting for differences in cell density (values normalized by chlorophyll fluorescence signal) (Figs 2C and 2D).

## Expression and secretion of industrial enzyme, xylanase, in *C. incerta*

For xylanase expression, *C. incerta* and *C. reinhardtii* were transformed with the pJP32Xyl vector in triplicate. For *C. incerta*, only 89 colonies were obtained for one of the three transformations with zero transformants for the other two samples. For *C. reinhardtii*, an average of 1604 colonies were obtained for each transformation (S2 Fig). Normalizing with the cell density using chlorophyll fluorescence, there was a statistically significant difference in the moles of product formed (μmol/s) between the two transgenic lines (adjusted p-value 0.00116) (Fig 3A). *C. incerta* demonstrated a higher percentage of positive transformants expressing xylanase, achieving 95.238% (80/84), compared to *C. reinhardtii*, which exhibited a lower rate of 66.667% (56/84) (Fig 3A). The threshold for positive expression was defined as three standard deviations above the mean activity level observed in the wild type strains. The three highest-expressing transgenic lines from each species were further evaluated for their ability

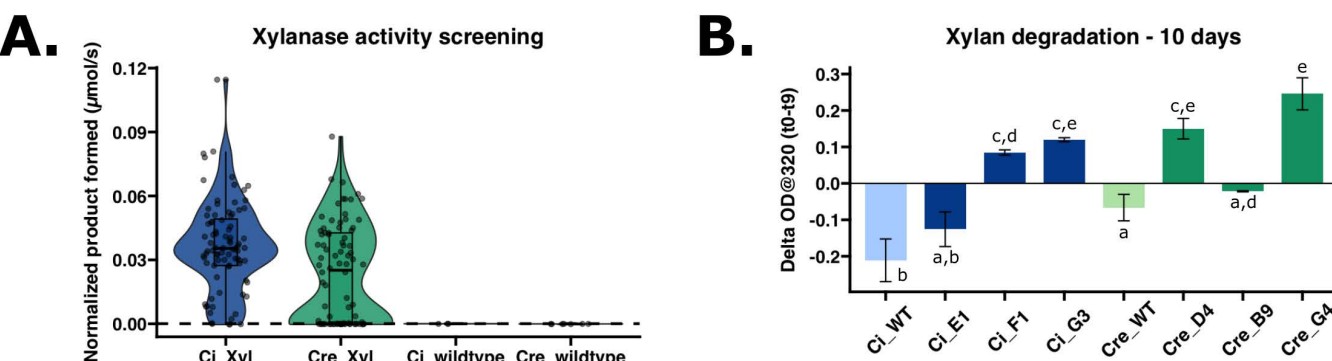

**Fig 3. Xylanase-expressing *C. incerta* and *C. reinhardtii* lines capable of fluorogenic substrate hydrolysis and commercial-grade xylan degradation. A)** Transgenic *C. incerta* (n = 84) and *C. reinhardtii* (n = 84) lines expressing xylanase demonstrated a statistically significant difference in activity by measuring the moles of product formed (μmol/s) over time (p-value 0.00116) via DiFMUX2 hydrolysis. Wild-type *C. incerta* (n = 3) and *C. reinhardtii* (n = 6) have no statistically significant difference (adjusted p-value 1.000) Fluorescence readings were measured at excitation wavelength of 355/9 nm and emission wavelength of 458/20 nm, and normalized with cell density using chlorophyll relative fluorescence unit (RFU). Threshold for positive expression (dashed line at $5.77 \times 10^{-5}$ μmol/s) was defined as three standard deviations above the mean activity level observed in the wild types. **B)** Two of the three highest xylanase-expressing lines from *C. incerta* (F1 and G3) and *C. reinhardtii* (D4 and G4) demonstrated the capability to degrade industrial-grade xylan after 9 days. Samples F1 and G3 from *C. incerta* exhibited a xylan degradation rate of 215.85 μg·min$^{-1}$·L$^{-1}$ and 305.06 μg·min$^{-1}$·L$^{-1}$, respectively. Samples D4 and G4 from *C. reinhardtii* exhibited a xylan degradation rate of 382.14 μg·min$^{-1}$·L$^{-1}$ and 626.02 μg·min$^{-1}$·L$^{-1}$, respectively.

to degrade xylan. While all transgenic lines showed activity in the EnzChek® Ultra Xylanase Activity Assay (S3 Fig), only two of the three highest-expressing lines from both *C. incerta* (F1 and G3) and *C. reinhardtii* (D4 and G4) demonstrated the capability to degrade industrial-grade xylan after 9 days (Fig 3B). Samples F1 and G3 from *C. incerta* exhibited a xylan degradation rate of 310.824 mg·d$^{-1}$·L$^{-1}$ and 439.286 mg·d$^{-1}$·L$^{-1}$, respectively. Comparably, samples D4 and G4 from *C. reinhardtii* exhibited a xylan degradation rate of 550.282 mg·d$^{-1}$·L$^{-1}$ and 901.469 mg·d$^{-1}$·L$^{-1}$, respectively. There was no statistically significant difference in xylan degradation rates, except between *C. incerta* strain F1 and *C. reinhardtii* strain G4 (adjusted p-value 0.0227 based on an ANOVA followed by a Tukey post-hoc analysis).

## Expression of plastic-degrading enzyme, PHL7, in *C. incerta*

The third type of recombinant protein tested was the plastic-degrading enzyme, PHL7 [32], expressed and secreted in *C. incerta* and *C. reinhardtii* transformed with the pJP32PHL7 vector [37]. Approximately 5,000 colonies were obtained per transformation event for *C. reinhardtii*, whereas *C. incerta* yielded only 0–10 colonies per plate across 12 transformations. Despite the low transformation efficiency in *C. incerta*, colonies from both species, along with their respective wild types, were picked into 96-well plates. To screen for positive transformants and confirm stable PHL7 expression, a replica of the 96-well plates was stamped onto a new TAP agar plate containing 0.5% (v/v) Impranil® DLN. After 7 days, 51 halos (51/81, 62.963%) were observed for *C. incerta* and 61 halos (61/84, 72.619%) for *C. reinhardtii* (Fig 4A). An Impranil® DLN-degrading activity assay was employed to further screen for PHL7 activity. Impranil® DLN, an opaque polyester polyurethane polymer suspension, becomes transparent upon degradation. Since PHL7 cleaves ester bonds in polymers, transformants secreting PHL7 formed transparent halos around the colonies on the opaque plates, indicating both secretion and activity of PHL7. This halo phenotype was leveraged to measure the decrease in absorbance as the enzyme degraded the polymer. The activity assay revealed a statistically significant difference in activity (change in OD at 350 nm over 14 days) between the two transgenic lines (adjusted p-value 4.2 x 10$^{-6}$) (Fig 4B). For *C. incerta* and *C. reinhardtii*, 50.617% (41/81) and 75% (63/84) of the colonies, respectively, displayed a negative

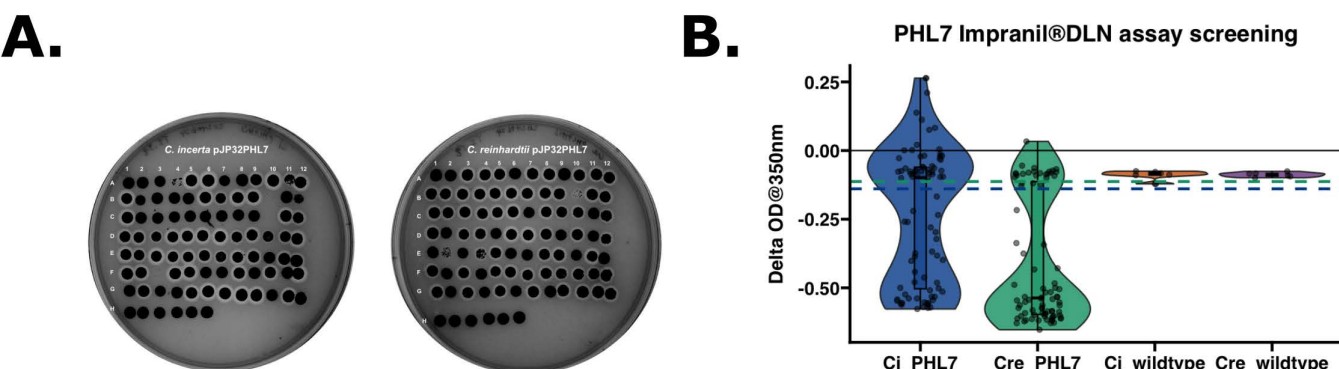

**Fig 4. Comparable PHL7 activity levels between *C. incerta* and *C. reinhardtii* based on Impranil® DLN degradation assays. A)** After 7 days, 52 halos (51/81, 62.963%) were observed for *C. incerta* (left) and 61 halos (61/84, 72.619%) for *C. reinhardtii* (right) on the TAP agar plates containing 0.5% (v/v) Impranil® DLN. For both replica plates, wells A1 to G12 represent potential transformants, wells H1 to H6 represent the respective wild types, and wells H7 to H12 represent blanks. **B)** The Impranil® DLN activity assay demonstrated a statistically significant difference in activity (i.e., change in OD at 350 nm over 14 days) between the transgenic *C. incerta* and *C. reinhardtii* lines (p-value 1.918 x 10−6). For *C. incerta* and *C. reinhardtii*, 23.457% (19/81) and 53.762% (46/84), respectively, of the colonies displayed a negative absorbance slope. Threshold for positive expression (blue dashed line at ΔOD -0.139 for *C. incerta* and green dashed line at ΔOD -0.113 for *C. reinhardtii*) was defined as three standard deviations below the mean activity level observed in the wild types.

absorbance slope, indicating esterase activity. The threshold for positive expression was defined as three standard deviations below the mean ΔOD observed in the wild type strains.

## Hybridization of *C. incerta* and *C. reinhardtii*

Sexual reproduction within species enhances phenotypic variation, increases genetic diversity, and offers evolutionary benefits, such as adaptation to environmental changes. It also allows for the exploitation of breeding programs to improve or incorporate desirable traits associated with a certain strain [38]. Since *C. incerta* demonstrated increased secretion of mCherry (and potentially other recombinant proteins), we thought it would be advantageous to hybridize *C. incerta* with another algal species, such as *C. reinhardtii*, which could allow for the development of new strains more tolerant to extreme environments, such as to saline environments (S4 Fig). However, because interspecies mating cannot occur between *C. incerta* and *C. reinhardtii* (S5 Fig) and mating types have yet to be established for *C. incerta*, hybridization by cell fusion of the two species were attempted.

The parent strains used for hybridization were transgenic *C. incerta* harboring a bleomycin resistance gene and transgenic *C. reinhardtii* (CC-620) harboring a *hygromycin* resistance gene. These transgenic lines were hybridized via electroporation, and the resulting progeny were selected on TAP agar plates containing 15 µg/mL zeocin and 30 µg/mL hygromycin B (Fig 5A). The growth of colonies on the double-selection plates demonstrated that the hybrid progeny successfully obtained both the zeocin and hygromycin resistance genes from their parents. The colonies were picked into 96-well plates, along with the parents and wild types, grown for 7 days, and stamped onto a new TAP agar plate with 15 µg/mL zeocin and 30 µg/mL hygromycin B to confirm that the new cell lines were stable (S6 Fig). Four hybrids that displayed resistance to zeocin and hygromycin were selected for further analyses (Fig 5B). Further confirmation of these four successful hybridizations was achieved through colony PCRs using oligonucleotides specific to the hygromycin B and bleomycin resistance genes (Figs 5C and 5D). Amplicons were obtained for only hybrids D11, D9, and C10. The hybrids resulting from cell fusion could be either haploid or diploid, as the fusion process does not guarantee the ploidy of the resulting progeny [39,40]. Nuclear fusion could have resulted in haploid hybrids, through genome shuffling, or diploid hybrids. Future ploidy analysis will be necessary to determine whether the ploidy of the progeny, which could influence their stability and phenotypic traits.

## Discussion

The results presented in this study highlight the promising potential of *C. incerta* as a novel host for the expression of recombinant proteins. Vectors originally designed for *C. reinhardtii* were utilized to express recombinant proteins in *C. incerta*, given their close phylogenetic relationship. As anticipated, several genetic elements, including the promoter, introns, and terminator, were recognized and functional in *C. incerta*, as demonstrated by resistance to antibiotic selection brought about by expression of the selectable marker genes for bleomycin and hygromycin, and by the robust accumulation of three different recombinant proteins. Furthermore, the functionality of the 2A peptide in *C. incerta* was validated by detecting unfused mCherry in various cellular compartments, including the cytosol, cell membrane, and cell wall via transformation using the pAH04mCherry, pJPM1mCherry, and pJPW2mCherry vectors, respectively. This confirmed the utility of the 2A peptide for polycistronic expression in this species. Of particular interest was the observation that the glycosylphosphatidylinositol (GPI) signal from *C. reinhardtii* was successfully recognized and lipidated in *C. incerta*. Evidence for this was the observation of successful membrane localization of mCherry to the

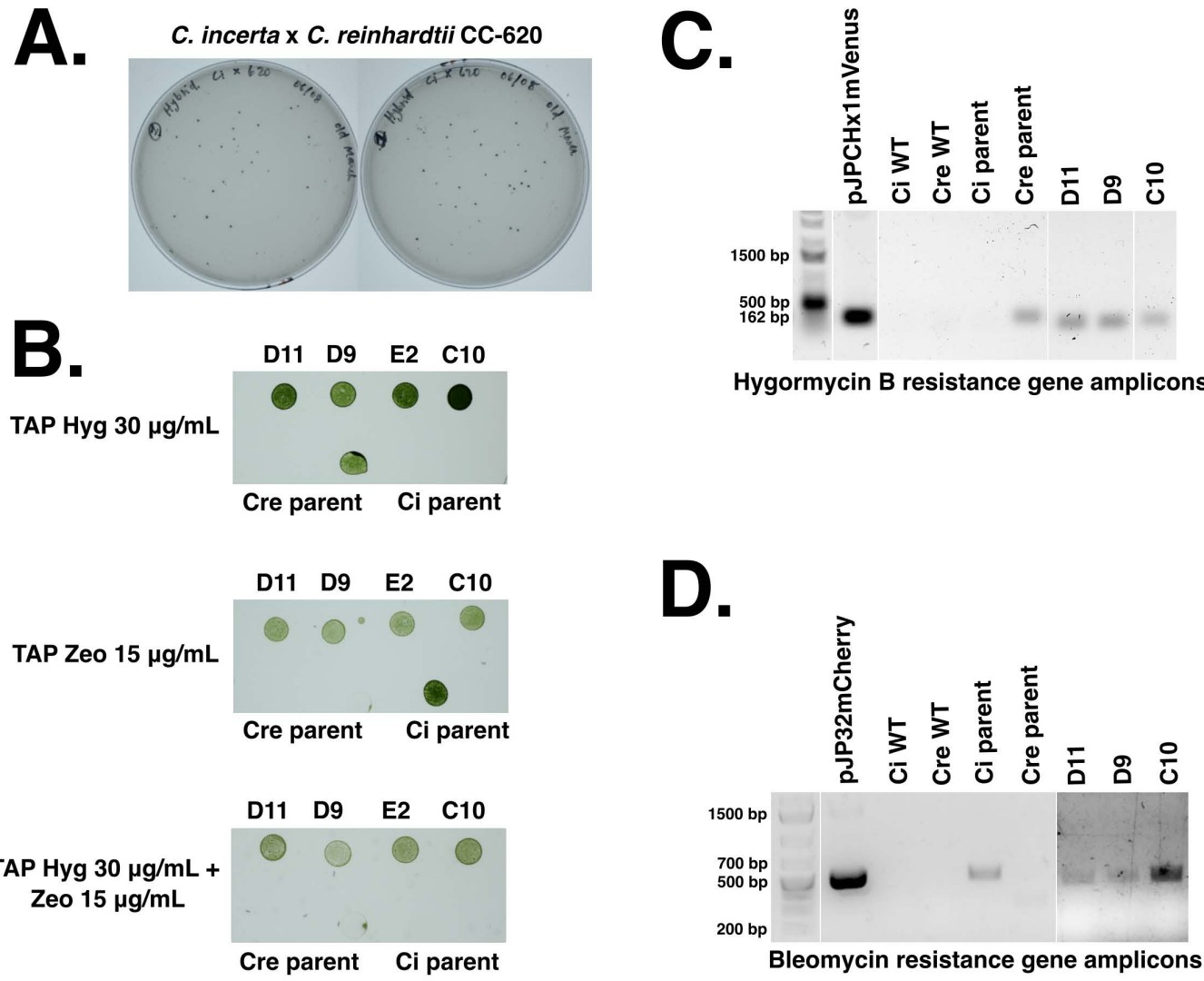

**Fig 5. Hybridizations of *C. incerta* holding bleomycin resistance and *C. reinhardtii* holding hygromycin B resistance. A)** Hybridizations were performed and 129 colonies were obtained on TAP agar plates containing 15 μg/mL zeocin and 30 μg/mL hygromycin. **B)** Four hybrids (D11, D9, E2, C10) that exhibited resistance to zeocin and hygromycin (and the parent strains) were plated onto TAP agar plates containing only 15 μg/mL zeocin, only 30 μg/mL hygromycin, or both 15 μg/mL zeocin and 30 μg/mL hygromycin. **C)** PCR analyses for the presence of bleomycin and hygromycin B resistance genes in the four hybrids, but amplicons for only 3 hybrids (D11, D9, C10) were obtained. The expected sizes of the PCR products were 500 bps for bleomycin and 162 bps for hygromycin B.

cellular membrane, thereby confirming the presence of GPI anchoring mechanisms in this species. This is a valuable tool to study the dynamic processes that constantly occur in the cell membrane, which was visualized in cells expressing mCherry in the membrane and flagella (Video 1). Additionally, microscopy demonstrated that the GP1-anchored protein GP1 from *C. reinhardtii* successfully anchored to the cell wall of *C. incerta*. Further evidence included recrystallization of the cell wall proteins via diafiltration. These findings exemplify the anticipated structural and functional similarities between the cell walls of these two species. Expression of mCherry in the cytosol, cell membrane, and cell wall were qualitatively comparable to expression of mCherry using the same vectors in *C. reinhardtii* [34]. Nevertheless, these findings indicated that *C. incerta* can be particularly valuable for applications requiring targeted

protein localization, such as intracellular metabolic engineering, motility-based applications, or enzymatic processes on the cell surface [41]. For instance, the ability of *C. incerta* to express functional proteins while retaining motility alludes to significant potential in motility-driven drug delivery, biosensing in dynamic environments, or the development of artificial micro-robots for biomedical and environmental purposes [2,42].

These findings also demonstrated that *C. incerta* exhibits notable expression and secretion capabilities compared to the well-studied *C. reinhardtii*, particularly for mCherry production. The approximately 3.5-fold (or 3.9-fold) higher mCherry fluorescence and 3.8-fold increase in immunoblot band intensity observed in *C. incerta* underscore its robust capacity for secretion of recombinant proteins into the extracellular medium. Such secretion of recombinant proteins into the medium is an essential feature for industrial applications in simplifying purification and other downstream processes.

In addition to mCherry, the ability of *C. incerta* to also express and secrete xylanase capable of degrading industrial-grade xylan holds substantial industrial significance. Xylanases are crucial enzymes in the bioconversion of lignocellulosic biomass into fermentable sugars, a process integral to several industrial sectors, including biofuel production, the pulp and paper industry, and textile industry [43–46]. This study revealed that *C. incerta* and *C. reinhardtii* showed comparable xylanase activity. However, to accurately determine the degradation capacity of *C. incerta*, further research is needed using methods capable of directly identifying and quantifying degradation products, such as gas chromatography-mass spectrometry (GC-MS) or high-performance liquid chromatography-mass spectrometry (HPLC-MS). Nevertheless, the ability of specific *C. incerta* transgenic lines to degrade industrial-grade xylan demonstrates its practical applicability in industrial processes. When comparing these rates to other systems, such as *Trichoderma spp.*, *Aspergillus spp.*, and *Penicillin spp.* [47–51], it becomes evident that making direct comparisons are challenging due to differences in units, assay conditions, and definitions of enzymatic activity. Despite this challenge, *C. incerta* presents significant advantages. As a microalga, it offers a more sustainable and potentially cost-effective platform for xylanase production compared to fungi. Microalgae can be cultivated in simple, nutrient-poor media using sunlight as an energy source, which can drastically reduce production costs [52,53].

To further investigate *C. incerta*'s capacity to express different proteins, a vector encoding the plastic-degrading enzyme PHL7 was also transformed into *C. incerta*. Interestingly, a low transformation efficiency was observed in *C. incerta* (less than 10 colonies per plate across 12 transformations) compared to *C. reinhardtii* (approximately 5,000 colonies per transformation). To a lesser extent, the low transformation efficiency was also observed when transforming with pJP32mCherry and pJP32Xylanase. This relatively low transformation efficiency in *C. incerta* may indicate differences in expression mechanisms or intrinsic genetic factors affecting transformation, such as repetitive elements and regulatory sequences. *C. incerta* has a larger genome (129.2 Mb) compared to *C. reinhardtii* (111.1 Mb), with the increase in genome size attributed to a higher content of repetitive elements [29]. Additionally, the centromeric regions in *C. incerta* are richer in repeats compared to *C. reinhardtii*, potentially influencing chromosomal stability and transformation efficiency, [29,54,55]. To better understand the evolutionary relationship between *C. incerta* and *C. reinhardtii*, a two-way average amino acid identity (AAI) analysis was performed, which revealed an AAI of 79.13%. This moderate level of protein sequence identity suggests that while the species are related, significant differences exist in their proteomes. Further protein-level comparisons of key genes support this conclusion. For instance, the ribulose-1,5-bisphosphate carboxylase/oxygenase small subunit (RbcS) from *C. incerta* had a 95.14% identity to that of *C. reinhardtii* (100% query coverage, E-value 1e-130), and the heat shock protein 70 (H70) had a 98.31% identity (100% query coverage,

E-value 0.0) (S1 Table). These high levels of identity suggest strong conservation in essential genes. However, other proteins like MAW8 (86.26% identity, 84% query coverage, E-value 1e-168) and GP1 (80.77% identity, 31% query coverage, E-value 5e-105) exhibited lower identity and coverage, indicating potential divergence in structural proteins like GP1 (S1 Table). Nonetheless, the similarities in their genomes allowed for similar gene expression using the RBCS2/HSP70 promoter. The MAW8 lipidation signal in *C. incerta* also retained sufficient similarity to allow for proper lipidation and processing, as confirmed by microscopy images (Fig 1D). Although the GP1 sequence exhibited low query coverage, likely due to the highly repetitive sequence of the gene, it was still sufficiently conserved to facilitate anchoring to the cell wall (Figs 1E–1H).

The transformation protocol that was optimized for *C. reinhardtii* may require modification for *C. incerta* to improve transformation efficiency. The stark difference in PHL7 expression could also be attributed to *C. incerta*'s lower tolerance to components of the Impranil suspension present in the TAP agar plate, further complicating the transformation process. All vectors used in this study contain selectable antibiotic resistance markers that were codon-optimized for *C. reinhardtii*. Analysis of *C. incerta* and *C. reinhardtii* revealed a correlation in codon usage patterns (S7 Fig). The similar codon usage patterns in both species suggest that DNA sequences optimized for *C. reinhardtii* are likely well-translated in *C. incerta*, suggesting that this is likely not a factor leading to lower apparent transformation efficiency. Ultimately, these findings underscore the need for species-specific protocols to enhance the success rate of recombinant protein expression. The low transformation efficiency in *C. incerta* could also suggest that not all species are equally suitable for all types of recombinant protein production, necessitating additional testing and optimization for each protein-species combination.

Despite the low transformation efficiency, a TAP agar 0.5% (v/v) Impranil® DLN replica plate of the 96-well plates containing the transformants and wild types visually demonstrated that PHL7 activity in *C. incerta* was comparable to that of *C. reinhardtii*. After 7 days, 51 clearing zones, or halos, were observed for *C. incerta* and 61 halos for *C. reinhardtii* (Fig 4A), indicating that both species were secreting active enzymes. An activity assay based on degradation of Impranil® DLN was performed and PHL7 expressed from *C. reinhardtii* exhibited higher activity, measured by a change in OD at 320 nm over 14 days, compared to that of *C. incerta* (Fig 4B). However, 41 out of 81 transformants for *C. incerta* and 63 out of 85 for *C. reinhardtii* displayed a negative absorbance slope—inconsistent with the number of positive transformants for both strains from the TAP agar 0.5% (v/v) Impranil® DLN replica plate. For *C. incerta*, the 96-well replica plate indicated that more transformants (specifically A11, B7, D2, D10, E2, E6, E7, E12, F5, and G2) exhibited PHL7 activity than otherwise observed in the activity assay. One factor that may contribute to this inconsistency is that the supernatant was collected and used for the Impranil® DLN activity assay, while the whole cells were present on the replica stamp plates which is continuously producing more PHL7 over time and thus accumulating more activity. In contrast, for *C. reinhardtii*, the activity assay indicated that two additional transformants (A9 and C7) exhibited PHL7 activity than otherwise observed on the 96-well replica plate. This inconsistency could have been attributed to the transgenic strains losing PHL7 secretion capacity, throught silencing, or some other mechanism. Nevertheless, the overall screening results were consistent with the findings for mCherry and xylanase in identifying true positive colonies, as well as previous studies involving *C. reinhardtii* [30].

This study illustrates that *C. incerta* can accumulate three distinct recombinant proteins, suggesting its promising application in biotechnology. However, these findings also indicate that *C. incerta* exhibits greater sensitivity to environmental conditions compared to *C. reinhardtii*. Notably, an environmental factor such as salinity can significantly influence its growth. Growth curves were established and *C. reinhardtii* thrived in TAP media containing

100 mM sodium chloride, while *C. incerta* merely tolerated this salt concentration (S4 Fig). This sensitivity underscores the importance of optimizing cultivation parameters to ensure consistent and reliable expression of recombinant proteins. To overcome this limitation of *C. incerta*, a potential solution is the use of breeding techniques to improve such traits, including mutagenesis, mating, and hybridization [40,56]. However, interspecies mating between *C. incerta* and *C. reinhardtii* is not feasible (S5 Fig), and mating types have yet to be characterized for *C. incerta*, necessitating the use of cell fusion hybridization techniques instead.

Intraspecies sexual reproduction enhances phenotypic diversity, augments genetic variation, and confers evolutionary advantages such as adaptation to changing environments. Given *C. incerta* has demonstrated capability for enhanced secretion of mCherry and potentially other proteins, hybridization with species like *C. reinhardtii*, which possesses traits such as higher saline tolerance, holds promise. Despite the impossibility of interspecies mating between *C. incerta* and *C. reinhardtii*, successful hybridizations, as demonstrated in this study, underscore the potential for significant advancements in algal biotechnology, including the combination of desirable traits from other algal species [40]. Additionally, hybridizing *C. incerta* with extremophiles could yield strains capable of producing recombinant proteins and thriving in extreme environments such as high pH and salinity—crucial for large-scale cultivation in open raceway ponds [57]. Hybridization of *C. incerta* and *C reinhardtii* was possible and was demonstrated by generating progeny resistant to both antibiotic resistance markers: bleomycin and hygromycin resistance genes, a trait obtained from both parents. The presence of these genes was confirmed by both selection on agar plates as well as identification of the specific genes using gene specific PCR. These results indicate the possibility of hybridization of both species, increasing their genetic diversity. Further research can now be pursued to improve traits in the hybrids, or backcross the hybrids with *C. reinhardtii,* further increasing the gene pool for *C. reinhardtii* technology.

## Conclusion

This study underscores *C. incerta* as a promising platform for recombinant protein production, showcasing its 3.5-fold higher mCherry expression and improved secretion capabilities compared to *C. reinhardtii*. The ability of *C. incerta* to effectively express mCherry in various cellular compartments: cytosol, cell membrane, and cell wall, and outside the cell, highlights its versatility and potential for applications requiring precise protein localization. Furthermore, the comparable xylanase activity between *C. incerta* and *C. reinhardtii* supports the utility of *C. incerta* in industrial processes, such as biofuel production and bioconversion of lignocellulosic biomass. The successful expression and activity of the plastic degrading enzyme PHL7 further emphasize *C. incerta*'s applicability in biotechnological endeavors, particularly in the remediation of plastic pollution. Additionally, the exploration of hybridization between *C. incerta* and *C. reinhardtii* opens new avenues for enhancing phenotypic diversity and developing strains with desirable traits for large-scale cultivation. Ultimately, these findings position *C. incerta* as a valuable host for recombinant protein expression, with significant implications for biotechnology, including sustainable production processes, environmental remediation, and the development of advanced biomedical applications.

## Materials and methods

### Construction of plasmids

All vectors were assembled using the pBlueScript II KS+ (pBSII) backbone. In the pJP32, pAH04, pJPM1, and pJPW2 vectors [30], the nuclear promoter AR1 (HSP70/RBCS2) was used. The *ble* gene was employed as the selection marker. The gene of interest was fused to

*ble* with a foot-and-mouth-disease-virus 2A self-cleavage peptide (FMDV-2A). For the pJP32, pJPM1 and pJPW2 vectors, *ble* is fused to FMDV-2A and a signal peptide, either SP7 or ARS1 signal peptide for secretion [30]. For the pAH04 vector, a signal peptide was not utilized to localize mCherry to the cytosol.

The pJP32mCherry vector [30] (also denoted as pJP32mCheHis) contains the mCherry sequence codon-optimized for *C. reinhardtii* that was purchased from IDT (Integrated DNA Technologies, San Diego, CA, USA). The pJP32Xylanase vector (pJP32XylHis) contains the xylanase sequence prepared by PCR from the pBR9_BD12 vector [31]. The pJP32PHL7 vector [37] (pJP32PHL7His) contains the PHL7 sequence codon-optimized for *C. reinhardtii* that was purchased from IDT. All pJP32 vectors also contain a polyhistidine-tag (6xHis-tag) at the C-terminus of all proteins of interest. The pAH04 vector contains the mCherry sequence codon-optimized for *C. reinhardtii* described in Molino et al., 2018 [30]. In the pJPW2 vector [34], SP7 and mCherry were linked to GP1, a hydroxyproline-rich protein hypothesized to be attached to the cell wall by non-covalent interaction [36], via a glycine-serine linker (GGGS-GGGS) and a 6xHis-tag. In the pJPM1 vector, the SP2 ARS1 was fused to mCherry. mCherry was fused to the last 49 amino acids of MAW8, a lipid-anchored protein that contains a putative GP1 anchoring signal [34]. The pJPCHx1 vector was assembled using genetic elements from the assembled genome from *Chlamydomonas pacifica* [57] and is described in Dutra Molino et al., 2024 [57]. Importantly, this vector contains the resistance marker to hygromycin, which was exploited to screen for hybrids.

All vector pieces were assembled using NEBuilder® HiFi DNA Assembly (NEB - New England Biolabs) following the manufacturer's protocol. All vectors contain restriction sites that flank the expression cassette (XbaI on the 5' end and KpnI on the 3' end) for linearization. The restriction enzymes were obtained from New England Biolabs (Ipswich, MA, USA). The final sequences can be found on Zenodo [58]. All vector maps can be found in S8, S9, and S10 Figs.

## C. incerta and C. reinhardtii growth conditions and transformation

**Growth conditions.** The wildtype, cell wall-containing *C. incerta* CC-3871, *C. reinhardtii* CC-1690 (mt+), *C. reinhardtii* CC-620 (mt+), and *C. reinhardtii* CC-621 (mt-) strains were obtained from the Chlamydomonas Resource Center in St. Paul, MN, USA. The strains were propagated in 50 mL TAP medium [59], using Kropat micronutrients recipe [60] at 25°C with constant illumination at 80 µmol photons $m^{-2}s^{-1}$ and agitated at 150 rpm on a rotary shaker.

**Transformation.** The plasmids were doubly digested using XbaI and KpnI restriction enzymes (New England Biolabs, Ipswich, MA, USA), followed by purification with the Wizard SV Gel and PCR Clean-up System (Promega Corporation, Madison, WI, USA) without fragment separation. DNA concentration was quantified using the Qubit dsDNA High Sensitivity Kit (Thermo Fisher Scientific, Waltham, MA, USA). The digested plasmids were transformed into *C. incerta* CC-3871, *C. reinhardtii* CC-1690, and *C. reinhardtii* CC-620 using electroporation. The cells were cultured to reach mid-logarithmic growth phase ($3 \times 10^6$–$6 \times 10^6$ cells/mL) in tris-acetate-phosphate (TAP) medium and maintained at 25°C with constant light exposure (80 µmol photons $m^{-2}s^{-1}$) on a rotary shaker at 150 rpm. The cells were concentrated by centrifugation at 3000 G for 10 minutes and resuspended to density of $3 \times 10^8$–$6 \times 10^8$ cells/mL using MAX Efficiency™ Transformation Reagent for Algae (Catalog No. A24229, Thermo Fisher Scientific). For electroporation, 250 µL of the cell suspension and 500 ng of a vector plasmid (digested with XbaI and KpnI restriction enzymes) was chilled on ice for five to ten minutes in a 4-mm wide cuvette compatible with the Gene Pulser®/MicroPulser™ (BioRad, Hercules, CA). The cell-DNA mixture was electroporated

using the GenePulser XCell™ device (BioRad, Hercules, CA) with a pulse of 2000 V/cm for 20 microseconds. Following electroporation, the cells were transferred into 10 mL of TAP medium and incubated with gentle agitation (50 rpm) in ambient light conditions (approximately 8 μmol photons m$^{-2}$s$^{-1}$) for an 18-hour recovery period. Post-recovery, the cells were concentrated by centrifugation, resuspended in 600 μL of TAP medium, and evenly spread onto two TAP agar plates with 15 μg/mL zeocin for pJP32mCherry- and pJP32Xylanase-transformed cells and two TAP agar plates with 15 μg/mL zeocin and 0.75% (v/v) Impranil® DLN for pJP32PHL7-transformed cells. The plates were incubated under a light intensity of 80 μmol photons m$^{-2}$s$^{-1}$ at 25 °C until visible colonies were formed. This transformation protocol can be found at [61].

## mCherry fluorescence analysis, growth curves, microscopy and western blotting

**mCherry fluorescence screening.** Transformant colonies were picked into 96-well plates: 84 transformant colonies, 6 wildtype *C. incerta* CC-3871 or *C. reinhardtii* CC-1690 colonies, and 6 blank wells. Each well contained 160 μL of TAP media. After a growth period of 7 days, fluorescence measurements were conducted using the Infinite® M200 PRO plate reader (Tecan, Männedorf, Switzerland). Chlorophyll was measured at an excitation wavelength of 440/9 nm and emission wavelength of 680/20 nm for cell density normalization. mCherry fluorescence was measured at an excitation wavelength of 580/9 nm and emission wavelength of 610/20 nm for analyzing protein secretion and expression. For mCherry fluorescence screening, a gain of 200 was used (Fig 1A). For the mCherry expression curves, a gain of 120 was used (Fig 1E). Transformants exhibiting the highest mCherry-to-chlorophyll ratio were selected for further analysis (growth curve, microscopy, and immunoblotting).

**Growth curves and expression curves.** Growth curves of the highest normalized mCherry fluorescence signal candidates from *C. incerta*, *C. reinhardtii*, and the wild types were established following the protocol from [62]. The growth curves represent the average of three biological replicates with three technical replicates. The appropriate volume of cells from a culture at stationary phase in TAP media was used to inoculate 6-well plates (GenClone, Genesee Scientific, El Cajon, CA, USA) containing 3 mL of TAP media to obtain an initial OD value (at 750 nm) of approximately 0.1 across all lines. The absorbances were measured at 750 nm everyday for 5 days using the Infinite® M200 PRO plate reader (Tecan, Männedorf, Switzerland). For the expression curves, mCherry fluorescence was measured at an excitation wavelength of 580/9 nm and emission wavelength of 610/20 nm for analyzing protein secretion and expression. A gain of 120 was used.

**Western blot.** Supernatants containing total secreted soluble protein were collected from three biological replicates of the highest expressing clones and wild types from *C. incerta* and *C. reinhardtii*. These replicates were pooled together and each were equally concentrated 60X using 10 kDa centrifugal filters (Amicon® Ultra, Darmstadt, Germany). The samples were denatured by adding 4X Laemmli Sample Buffer (BioRad, Hercules, CA), followed by boiling at 98°C for 10 minutes. Equal volumetric amounts of protein samples (30 μL) were loaded into the 12% TGX Stain-Free™ FastCast™ gel (BioRad, Hercules, CA). The proteins were separated at 80 V to 120 V. After separation, analysis of the Stain-Free™ gel using ImageJ (Schindelin et al., 2012) conveyed that roughly equal amounts of total soluble protein were loaded for the transgenic lines (16540.04 and 13975.85 arbitrary units for *C. incerta* and *C. reinhardtii*, respectively). For *C. incerta* and *C. reinhardtii* wild types, 123817.79 and 9447.28 arbitrary units, respectively, were loaded. The values for total soluble protein loaded in the gel were calculated using Stain-Free Total Protein Quantification method described here: (Hammond,

2021). The proteins were then transferred to a nitrocellulose membrane at 15 V for 1 hour. The membrane was blocked with 1% Bovine Serum Albumin (Sigma-Aldrich, St. Louis, MO, USA) diluted in 1X phosphate-buffered saline with 0.1% Tween® detergent overnight at 4°C on a rocking shaker (BioRocker™ 2D Rockers, Thermo Fisher Scientific, Waltham, MA, USA). After blocking, the membrane was probed with an anti-RFP polyclonal antibody conjugated to horse radish peroxidase (Abcam, Cambridge, United Kingdom) for 1 hour at room temperature on a rocking shaker (BioRocker™ 2D Rockers, Thermo Fisher Scientific, Waltham, MA, USA). Bands were detected using Clarity Western ECL Blotting Substrate (BioRad, Hercules, CA) as per the manufacturer's instructions.

## Cellular fluorescence localization

Transformed strains were cultivated in TAP medium until they reached the late log phase at 25°C, under continuous illumination of 80 µmol photons/m²/s, with agitation at 150 rpm on a rotary shaker. Live cells were then observed using agarose pads, prepared according to the protocol described at [62]. These cells were placed onto TAP 1% agarose pads, prepared with Frame-Seal™ Slide Chambers (15 × 15 mm, 65 µL) on a glass slide and covered with a cover-slip before image acquisition. Live-cell imaging was conducted using an automated inverted confocal laser scanning microscope (Leica Stellaris 5 Confocal). mCherry fluorescence was excited at 580 nm with a laser set to 5% power, and emission was detected using a HyD hybrid detector set between 601 nm and 634 nm. Chlorophyll fluorescence was excited at 405 nm with a laser set to 2% power, and emission was detected between 650 nm and 750 nm, again using the HyD hybrid detector. Microscope settings were maintained consistently across all imaging sets, and images were acquired in sequence mode for both chlorophyll and mCherry. Image analysis was performed using Fiji [63], an ImageJ distribution, with uniform settings applied across all image groups. Brightness adjustments were made in Fiji, with consistent settings used unless otherwise noted. For video imaging, a Zeiss Elyra 7 Lattice SIM microscope was used. In this setup, a diode-pumped solid-state laser with a 561 nm wavelength was used to excite mCherry, with BP 525/50 and BP 617/73 filters in line with a long-pass dichroic mirror SBS LP 560 to observe mCherry fluorescence.

## Chemical extraction of cell wall components and recrystallization of mCherry

This protocol was adapted from Goodenough et al. to extract the chaotropic soluble portion of the *C. reinhardtii* cell wall [35]. Strains were cultured for 5 days, and 1 mL of the culture was centrifuged at 3000 g for 2 minutes. The pellet was washed with 1 mL of ddH₂O, followed by centrifugation. The cell pellet was then resuspended in 150 µL of 2 M sodium perchlorate, centrifuged at 20000 g for 1 minute, and 100 µL of the supernatant was used for fluorescence reading at excitation wavelength of 580/9 nm and emission wavelength of 610/20 nm.

For cell wall recrystallization, 40 mL of culture was centrifuged at 3000 g for 3 minutes. The pellet was washed with 40 mL of ddH₂O and resuspended in 1 mL of 2 M sodium per-chlorate. After transferring to a microcentrifuge tube, the mixture was centrifuged at 20000 g for 1 minute. Approximately 900 µL of the supernatant was concentrated to 50 µL using a 30 K centrifugal filter (Amicon® Ultra, Darmstadt, Germany), followed by three diafiltration steps with 450 µL ddH₂O. The cell wall crystals were recovered according to the manufacturer's instructions by inverting the filter membrane into a new collection tube. Cell wall crystal images were acquired using the ChemiDoc™ MP Imaging System (Bio-Rad, Hercules, CA). Crystal proteins were detected via the Stain-Free method, which employs a UV light source to excite aromatic amino acid residues, allowing for direct visualization of proteins without the

need for traditional staining. For Cy3 detection, the system utilizes a green LED light source (520–550 nm) to excite the mCherry fluorophore, with an emission filter (575–610 nm) ensuring precise fluorescent signal detection.

## Xylanase activity assays and degradation of xylan

**Xylanase activity assay on supernatant samples.** *C. incerta* and *C. reinhardtii* transformant colonies were picked into two 96-well plates. Each plate contained 84 transformant colonies, 6 of their respective wildtype *C. incerta* CC-3871 or *C. reinhardtii* CC-1690 colonies, and 6 blank wells. Each well contained 160 μL of TAP media. After a growth period of 7 days, fluorescence measurements were conducted using the Infinite® M200 PRO plate reader (Tecan, Männedorf, Switzerland). Chlorophyll was measured at an excitation wavelength of 440/9 nm and emission wavelength of 680/20 nm for cell density normalization. The 96-well plates were centrifuged at 2000 G for 5 minutes, and the supernatant was collected and filtered through a 0.2 μm PVDF Membrane 96-well filter plate (Corning®, Corning, NY, USA). The filtered supernatant (total soluble protein) was collected in a 96-well PCR plate (Fisher Scientific, Waltham, MA, USA) from which 50 μL was taken for the xylanase activity assay. Xylanase activity was measured using the EnzChek® Ultra Xylanase Activity Kit (Life Technologies, Carlsbad, CA). Hydrolysis of the fluorogenic substrate 6,8-difluoro-4-methylumbelliferyl β-d-xylobioside (DiFMUX2) by xylanase led to increased fluorescence at an excitation wavelength of 385 nm and emission wavelength of 455 nm over time [64]. The activity of algal-expressed xylanase was compared to 25 mU (1.25 μg) commercial Xylanase from *Trichoderma longibrachiatum* (Sigma-Aldrich, St. Louis, MO, USA). 50 μL of xylanase substrate (12.5 μg) was added to 50 μL of xylanase-containing samples in a clear 96-well flat-bottom well plate (GenClone, Genesee Scientific, El Cajon, CA, USA). Fluorescence readings were measured at excitation wavelength of 355/9 nm and emission wavelength of 458/20 nm every 4 minutes for approximately 32 minutes, incubated at 42°C using the Infinite® M200 plate reader (Tecan, Männedorf, Switzerland). Product formation rates (μmol/s) were calculated as per the manufacturer's instructions.

**Degradation of xylan.** Three samples with the highest normalized product formed ((μmol/s)/chlorophyll RFU) and the wildtypes from *C. incerta* and *C. reinhardtii* were selected, expanded to 3 mL cultures in TAP media in 6-well plates (GenClone, Genesee Scientific, El Cajon, CA, USA), and grown for 5 days at 25°C with constant light exposure (80 μmol photons m$^{-2}$s$^{-1}$) on a rotary shaker at 150 rpm. The xylanase-containing supernatants were collected and concentrated 40X using 10 kDa centrifugal filters (Amicon® Ultra, Darmstadt, Germany). In 0.2 mL PCR tubes (Olympus Plastics, Genesee Scientific, El Cajon, CA, USA), 60 μL of sodium acetate buffer (1M, pH 4.5) with 1% Xylan from Corn Core (Tokyo Chemical Industry, Tokyo, Japan) and 140 μL of the concentrated xylanase-containing samples were mixed, totalling 200 μL with a final 28X-concentrated xylanase sample. Absorbance readings at 320 nm were taken on the day the samples were prepared (time point 0) and after 9 days (time point 10) in 96-well UV-Star® microplates (Greiner Bio-One, Kremsmünster, Austria) using the Infinite® M200 PRO plate reader (Tecan, Männedorf, Switzerland). The degradation represents the average of two biological replicates with two technical replicates.

The xylan absorbance standard curve was created using varying concentrations of Xylan from Corn Core (0%, 0.0625%, 0.125%, 0.25%, 0.5%, 1%, and 2% w/v) in sodium acetate buffer (1M, pH 4.5) (S11 Fig). Absorbance at 320 nm were measured in the 96-well UV-Star® microplates (Greiner Bio-One, Kremsmünster, Austria) using the Infinite® M200 PRO plate reader (Tecan, Männedorf, Switzerland).

## Impranil® DLN activity assay

*C. incerta* and *C. reinhardtii* transformant colonies were picked into two 96-well plates. Each plate contained 84 transformant colonies, 6 of their respective wildtype *C. incerta* CC-3871 or *C. reinhardtii* CC-1690 colonies, and 6 blank wells. Each well contained 160 μL of TAP media. After a growth period of 7 days, fluorescence measurements were conducted using the Infinite® M200 PRO plate reader (Tecan, Männedorf, Switzerland). Chlorophyll was measured at an excitation wavelength of 440/9 nm and emission wavelength of 680/20 nm for cell density normalization. The 96-well plates were centrifuged at 2000 G for 5 minutes, and the supernatant was collected and filtered through a 0.2 μm PVDF Membrane 96-well filter plate (Corning®, Corning, NY, USA). The filtered supernatant (total soluble protein) was collected in a 96-well PCR plate (Fisher Scientific, Waltham, MA, USA) from which 50 μL was taken for the polyester polyurethane polymer degradation assay.

This protocol was adapted from [37]. The 96-well round bottom plate (Thermo Fisher Scientific, Waltham, MA, USA) was prepared by heating a mixture of 0.2% (w/v) agarose and 1 M $K_2HPO_4$ buffer solution in a 50 ml erlenmeyer flask in a microwave until the agarose is completely melted. Impranil® DLN, a colloidal polyester-PUR dispersion [65], was then added to the hot solution to reach a final concentration of 0.25% (v/v) and mixed thoroughly to ensure homogenous distribution. Subsequently, 150 μL of the molten Impranil® DLN-agarose solution was pipetted into each well of a 96-well round bottom plate and allowed to solidify at room temperature. Once the gel had solidified, 50 μL of the enzyme preparation was pipetted into each well. The assay plate was tightly sealed using a transparent 96 Well Plate Sealing Film (Lichen Cottage, Item Number SF-100). Using the Infinite® M200 PRO plate reader (Tecan, Männedorf, Switzerland), absorbance readings were measured at 350 nm, every 5 minutes, for over the course of 14 days. PHL7 activity was analyzed by calculating the change in absorbance over time.

## Mating *C. reinhardtii* CC-620 (mt+) and *C. reinhardtii* CC-621 (mt-)

*C. reinhardtii* CC-620 (mt+) and *C. reinhardtii* CC-621 (mt-) were grown until stationary phase in TAP media at 25°C with constant light exposure (80 μmol photons $m^{-2}s^{-1}$) on a rotary shaker at 150 rpm. On separate TAP media agar plates, 1 mL of the dense cell culture was pipetted and spread across the entire surface of the plate. The plates were incubated under a light intensity of 80 μmol photons $m^{-2}s^{-1}$ at 25 °C until a lawn of cells formed. For each plate, the cells were scraped and transferred to 10 mL of TAP media without nitrogen (TAP-N) in a sterile 50 mL centrifuge tube. Cell clumps were resuspended and pipetted into a sterile 100 mm x 15 mm petri dish (Falcon, Corning, NY, USA). The plates were incubated under 80 μmol photons $m^{-2}s^{-1}$ of light at 25 °C without shaking for 6–8 hours. The cultures were then collected and combined into a new sterile 150 mm x 15 mm petri dish (Fisher Scientific, Waltham, MA, USA). The plate was incubated in ambient light conditions (approximately 8 μmol photons $m^{-2}s^{-1}$) at room temperature without shaking overnight (12–14 hours). Strains were successfully mated once a pellicle layer at the bottom of the petri dish was observed the next day. This protocol was adapted from [66].

## Hybridization

**Autolysin production.** *C. reinhardtii* CC-620 (mt+) and *C. reinhardtii* CC-621 (mt-) were mated following the protocol described above. Once a pellicle layer at the bottom of the petri dish was observed the next day, the supernatant was collected, filtered (0.45 μm 30 mm diameter syringe filter [GenClone, Genesee Scientific, El Cajon, CA, USA]), and used for hybridization. This protocol was adapted from [66].

**Hybridization using electroporation.** The transgenic *C. incerta* CC-3871 strain (pJP32mChe) with zeocin resistance, and the transgenic *C. reinhardtii* CC-620 (mt+) strain (pJPCHmVen) with hygromycin resistance were selected for hybridization. Both transgenic strains were grown until stationary phase in TAP media at 25°C with constant light exposure (80 µmol photons m$^{-2}$s$^{-1}$) on a rotary shaker at 150 rpm. At stationary phase, 20 mL of each strain pipetted into sterile 50 mL centrifuge tubes and centrifuged at 3000 G for 10 minutes. The cell pellets were then washed with 20 mL of TAP media, and centrifuged at 3000 G for 10 minutes. The cell pellets were resuspended in 20 mL of autolysin and incubated in ambient light conditions (approximately 8 µmol photons m$^{-2}$s$^{-1}$) at room temperature without shaking for 2–4 hours. After this incubation, the two transgenic strains were mixed together in a new sterile 50 mL centrifuge tube. The mixed cells were then concentrated by centrifugation at 3000 G for 10 minutes and resuspended to density of 3 x 10$^8$–6 x 10$^8$ cells/mL using MAX Efficiency™ Transformation Reagent for Algae (Catalog No. A24229, Thermo Fisher Scientific). For electroporation, 250 µL of the mixed cell suspension was chilled on ice for 5–10 minutes in a 4-mm wide cuvette compatible with the Gene Pulser®/MicroPulser™ (BioRad, Hercules, CA). The cell mixture was electroporated using the GenePulser XCell™ device (BioRad, Hercules, CA) with a pulse of 2000 V/cm for 20 microseconds. Following electroporation, the cells were transferred into 10 mL of TAP medium and incubated with gentle agitation (50 rpm) in ambient light conditions (approximately 8 µmol photons m$^{-2}$s$^{-1}$) for an 18-hour recovery period. Post-recovery, the cells were concentrated by centrifugation, resuspended in 400 µL of TAP medium, and evenly spread onto two TAP agar plates with the 15 µg/mL zeocin and 30 µg/mL hygromycin. The plates were incubated under a light intensity of 80 µmol photons m$^{-2}$s$^{-1}$ at 25 °C until visible colonies were formed.

**PCR screens for successful hybridization.** The successful hybridization of the transgenic *C. incerta* pJP32mChe and transgenic *C. reinhardtii* pJPCHmVen was determined by colony PCR. Cell cultures of the hybrids were boiled at 98°C for 10 minutes, and the cell lysate was used as the template for the reaction. The bleomycin and hygromycin B genes that are fused to mCherry and mVenus, respectively, were amplified. For the *ble* gene positive screens, the oligonucleotides'5'- CGGGTTCTCCCGGGACTTCG - 3' and 5' ACCTCCGACCACTCGGCGTA -3' were used. For the hygromycin B gene positive screens, the oligonucleotides 5' - TGATTCCTACGCGAGCCTGC - 3' and 5' - AACAGCTTGATCACCGGGCC - 3' were used. The thermal cycler settings consisted of 98°C for 45 seconds, 65°C for 30 seconds, 72°C for 2 minutes, and an extension of 72°C for 2 minutes. Both genes were amplified for 30 cycles. The PCR reactions were set up with Q5 Master Mix (New England Biolabs, Ipswich, MA, USA) according to the manufacturer's protocol.

## Data analysis

R Statistic version 3.6.3 (2020-02-29) running in the RStudio v1.2.5042 IDE was used to import and generate plots for the figures and the ANOVAs followed by Tukey post-hoc analyses. The ggprism theme was used to generate all graphs and plots [67]. All data were analyzed and processed using Microsoft Excel 365 (Version 16.0, Microsoft Corporation, Redmond, WA, USA).

## Supporting information

**S1 Fig. Transformations of pJP32mCherry into *C. reinhardtii* and *C. incerta*.** The mCherry secretion vector, pJP32mCherry, was transformed into **A)** *C. reinhardtii* and **B)** *C. incerta*. Triplicate transformations were performed for both species, and transformants were spread

onto two plates (paired vertically). However, only two transformations for *C. incerta* generated an adequate amount of colonies, so the third transformation is not shown.
(TIF)

**S2 Fig. Transformations of pJP32Xylanase into *C. reinhardtii* and *C. incerta*.** The xylanase secretion vector, pJP32Xylanase, was transformed into **A)** *C. reinhardtii* and **B)** *C. incerta*. Triplicate transformations were performed for both species, and transformants were spread onto two plates (paired vertically). However, only one transformation for *C. incerta* generated an adequate amount of colonies, so the other two transformations are not shown.
(TIF)

**S3 Fig. Xylanase activity assay of three highest expressing *C. incerta* and *C. reinhardtii* transgenic lines of pJP32Xylanase.** Hydrolysis of the fluorogenic substrate 6,8-difluoro-4-methylumbelliferyl β-d-xylobioside (DiFMUX2) by xylanase led to increased fluorescence at an excitation wavelength of 385 nm and emission wavelength of 455 nm over time. The F1, E1, and G3 strains of transgenic *C. incerta* expressing xylanase formed 302.799 μmol/s, 239.806 μmol/s, and 272.852 μmol/s of product, respectively. The D4, B9, and G3 strains of transgenic *C. reinhardtii* expressing xylanase formed 188.873 μmol/s, 235.523 μmol/s, and 163.695 μmol/s of product, respectively. The *C. incerta* and *C. reinhardtii* wild types formed 2.219 μmol/s and 1.918 μmol/s of product, respectively.
(TIF)

**S4 Fig. Growth curves of *C. incerta* and *C. reinhardtii* in TAP media with NaCl. A)** The *C. incerta* wild type was grown in TAP media containing 0 mM, 100 mM, 200 mM, 300 mM, 400 mM, and 500 mM NaCl. Absorbance readings at 750 nm were measured using the Infinite® M200 PRO plate reader (Tecan, Männedorf, Switzerland) over approximately 73 hours, and the readings represent the average of biological quadruplicates. B) The *C. reinhardtii* wild type was grown in TAP media containing 0 mM, 100 mM, 200 mM, 300 mM, 400 mM, and 500 mM NaCl. Absorbance readings at 750 nm were measured using the Infinite® M200 PRO plate reader (Tecan, Männedorf, Switzerland) over approximately 73 hours, and the readings represent the average of biological quadruplicates.
(TIF)

**S5 Fig. Unsuccessful interspecies mating between *C. incerta* and *C. reinhardtii*. A)** *C. reinhardtii* CC-620 (mt+) wild type and C. *reinhardtii* CC-621 (mt-) wild type were mated, and a pellicle phenotype was observed after 12–16 hours, indicating success of intraspecies mating. **B)** Transgenic *C. reinhardtii* CC-620 (mt+) pJPCHx1mVenus and transgenic *C. incerta* pJP32mCherry, and a pellicle phenotype was not observed after 12–16 hours, indicating that interspecies mating is not feasible. C) The mixed cells that did not formed a pellicle from B, along with the *C. reinhardtii* pJPCHx1mVenus and *C. incerta* pJP32 parents were plated onto 3 types of plates: TAP agar plates containing zeocin 15 μg/mL, TAP agar plates containing hygromycin B 30 μg/mL, and TAP agar plates containing both zeocin 15 μg/mL and hygromycin 30 μg/mL.
(TIF)

**S6 Fig. Screening for successful hybridizations between transgenic *C. incerta* pJP32mCherry and transgenic *C. reinhardtii* CC-620.** Potential hybrid colonies from the transformation plate were picked into 96-well plates containing TAP media and grown for 7 days. Replica TAP agar plate containing zeocin 15 μg/mL and hygromycin 30 μg/mL was made to screen for stable hybrids.
(TIF)

**S7 Fig. Codon usage comparison between *C. incerta* and *C. reinhardtii.*** The scatter plot depicts the frequency of each codon's usage in C. *incerta* (x-axis) against C. *reinhardtii* (y-axis). A positive 1:1 correlation in codon usage patterns is observed between the two species, suggesting similarities in their codon preferences despite genomic differences.
(TIF)

**S8 Fig. Plasmid maps for pJP32 secretion vectors.** Plasmid maps for **A)** pJP32mCheHis (denoted as pJP32mCherry in the publication), **B)** pJP32XylHis (pJP32Xylanase), and C) pJP-32PHL7His (pJP32PHL7) using SnapGene (GSL Biotech LLC, San Diego, CA, USA).
(TIF)

**S9 Fig. Plasmid maps for pAH04, pJPM1, and pJPW2 vectors.** Plasmid maps for **A)** pAH-04mCherry, an mCherry cytosolic expression vector, **B)** pJPM1mCherry, an mCherry cell membrane expression vector, and C) pJPW2mCherry, an mCherry cell wall expression vector, using SnapGene (GSL Biotech LLC, San Diego, CA, USA).
(TIF)

**S10 Fig. Plasmid map for pJPCHx1mVenus vector.** Plasmid map for pJPCHx1mVenus, an mVenus cytosolic expression vector containing the hygromycin B antibiotic resistance gene, using SnapGene (GSL Biotech LLC, San Diego, CA, USA).
(TIF)

**S11 Fig. Standard curve of absorbance of Xylan from Corn Core.** The xylan absorbance standard curve was created using varying concentrations of Xylan from Corn Core (0%, 0.0625%, 0.125%, 0.25%, 0.5%, 1%, and 2% w/v) in sodium acetate buffer (1M, pH 4.5). Absorbance at 320 nm were measured in the 96-well UV-Star® microplates (Greiner Bio-One, Kremsmünster, Austria) using the Infinite® M200 PRO plate reader (Tecan, Männedorf, Switzerland). The standard curve was fitted with a linear regression equation $y = mx + b$, where m is the slope and b is the y-intercept (equation: $y = 0.2727x + 0.0414$; $R2 = 0.9997$).
(TIF)

**S1 Table. BLASTp results comparing protein sequences of genes with DNA parts in the between *C. incerta* and reference protein sequences.** The table shows the GenBank IDs for the queried genes, the best hit GenBank protein ID, the E-value indicating the statistical significance of the match, the query coverage percentage, and the percentage identity of the aligned sequences.
(TIF)

## Acknowledgements

The authors gratefully acknowledge the support of imaging specialist Marcy Erb, Ph.D. for her contribution in the microscopy core.

## Author contributions

**Conceptualization:** Kalisa Kang, João Vitor Dutra Molino.

**Data curation:** Kalisa Kang, Aaron Oliver, Lauren May, João Vitor Dutra Molino.

**Formal analysis:** Kalisa Kang, João Vitor Dutra Molino.

**Funding acquisition:** Stephen Mayfield.

**Investigation:** Kalisa Kang, Évellin do Espirito Santo, Crisandra Jade Diaz, Lisa Saxton, Lauren May, João Vitor Dutra Molino.

**Methodology:** Kalisa Kang, João Vitor Dutra Molino.

**Project administration:** Kalisa Kang, João Vitor Dutra Molino.

**Software:** Aaron Oliver.

**Supervision:** Stephen Mayfield, João Vitor Dutra Molino.

**Visualization:** Kalisa Kang, João Vitor Dutra Molino.

**Writing – original draft:** Kalisa Kang, Stephen Mayfield, João Vitor Dutra Molino.

**Writing – review & editing:** Kalisa Kang, Évellin do Espirito Santo, Crisandra Jade Diaz, Aaron Oliver, Lisa Saxton, Lauren May, Stephen Mayfield, João Vitor Dutra Molino.

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
