## [Decision Letter · Decision Letter 0]

27 Jan 2025

PONE-D-24-49646Establishing the green algae Chlamydomonas incerta as a platform for recombinant protein productionPLOS ONE

Dear Dr. Molino,

Thank you for submitting your manuscript to PLOS ONE. After careful consideration, we feel that it has merit but does not fully meet PLOS ONE’s publication criteria as it currently stands. Therefore, we invite you to submit a revised version of the manuscript that addresses the points raised during the review process.

We look forward to receiving your revised manuscript.

Kind regards,

Andrew Webber

Academic Editor

PLOS ONE

**Journal Requirements:**

This material is based upon work supported by the U.S. Department of Energy’s Office of Energy Efficiency and Renewable Energy (EERE) under the APEX award number DE-EE0009671. This material was also supported by the S10OD030505 and NINDS P30NS047101 grants from the UCSD microscopy core.

The authors gratefully acknowledge the support from the United States Department of Energy, the California Center for Algae Biotechnology, and the University of California San Diego. They also acknowledge the support of imaging specialist Marcy Erb, Ph.D. for her contribution in the microscopy core. 

This material is based upon work supported by the U.S. Department of Energy’s Office of Energy Efficiency and Renewable Energy (EERE) under the APEX award number DE-EE0009671. This material was also supported by the S10OD030505 and NINDS P30NS047101 grants from the UCSD microscopy core.

Stephen Mayfield was a founding of and holds an equity stake in Algenesis Inc, a company that could potentially benefit from this work. The remaining authors declare that the research was conducted in the absence of any commercial or financial relationships that could be construed as a potential conflict of interest.

We note that one or more of the authors are employed by a commercial company: Algenesis Inc,. 

“The funder provided support in the form of salaries for authors, but did not have any additional role in the study design, data collection and analysis, decision to publish, or preparation of the manuscript. The specific roles of these authors are articulated in the ‘author contributions’ section.”

**Additional Editor Comments:**

I agree withe the reviewer and the easiest approach will be to revise the title. Please return the manuscript and include a detailed response of how you chose the new title.

Reviewers' comments:

Reviewer's Responses to Questions

**Comments to the Author**

1. Is the manuscript technically sound, and do the data support the conclusions?

Reviewer #1: Yes

2. Has the statistical analysis been performed appropriately and rigorously? 

Reviewer #1: Yes

3. Have the authors made all data underlying the findings in their manuscript fully available?

Reviewer #1: Yes

4. Is the manuscript presented in an intelligible fashion and written in standard English?

Reviewer #1: Yes

5. Review Comments to the Author

**Reviewer #1: ** The manuscript by Kang et al. describes strain engineering work comparing protein expression titers in C. incerta and C. reinhardtii. A 3-fold increase in transgene fluorescence and protein titer was shown for one gene construction in C. incerta relative to C. reinhardtii. Other constructs provided similar results between the strains. Protein targeting to various cellular compartments were also evaluated to demonstrate strain capabilities.

I have two related recommendations to provide the authors. The title of the manuscript states the objective of the work is "platform" development for recombinant protein production. That implies an entire production system from strain improvement to scaled up cultivation. Yet the work that is described covers gene design through phenotyping at a scale that is not described. Hence, recommendation one is to add detail to the cultivation section of the M&M to include volumes cultivated and culture conditions used; that would include sterility requirements. Second, please describe the intended cultivation platform that leads to economic viability. Would that involve raceway ponds, photobioreactors - and what type of each would be compatible with the strains described. What are the maximum culture volumes have been achieved with Chlamydomonas recombinant strains. All those details will be of great interest to those considering adoption and could easily be provided in the introduction or M&M sections. Alternative, the authors could elect to change the title to reflect the strain engineering focus of the work without reference to platform development.

6. PLOS authors have the option to publish the peer review history of their article (what does this mean? ). If published, this will include your full peer review and any attached files.

**Do you want your identity to be public for this peer review?** For information about this choice, including consent withdrawal, please see our Privacy Policy .

Reviewer #1: No

---

## [Author Response · Author response to Decision Letter 1]

27 Feb 2025

Reviewers' comments:

Reviewer's Responses to Questions

Comments to the Author

1. Is the manuscript technically sound, and do the data support the conclusions?

Reviewer #1: Yes

2. Has the statistical analysis been performed appropriately and rigorously?

Reviewer #1: Yes

3. Have the authors made all data underlying the findings in their manuscript fully available?

Reviewer #1: Yes

4. Is the manuscript presented in an intelligible fashion and written in standard English?

Reviewer #1: Yes

5. Review Comments to the Author

Reviewer #1: The manuscript by Kang et al. describes strain engineering work comparing protein expression titers in C. incerta and C. reinhardtii. A 3-fold increase in transgene fluorescence and protein titer was shown for one gene construction in C. incerta relative to C. reinhardtii. Other constructs provided similar results between the strains. Protein targeting to various cellular compartments were also evaluated to demonstrate strain capabilities.

I have two related recommendations to provide the authors. The title of the manuscript states the objective of the work is "platform" development for recombinant protein production. That implies an entire production system from strain improvement to scaled up cultivation. Yet the work that is described covers gene design through phenotyping at a scale that is not described. Hence, recommendation one is to add detail to the cultivation section of the M&M to include volumes cultivated and culture conditions used; that would include sterility requirements. Second, please describe the intended cultivation platform that leads to economic viability. Would that involve raceway ponds, photobioreactors - and what type of each would be compatible with the strains described. What are the maximum culture volumes have been achieved with Chlamydomonas recombinant strains. All those details will be of great interest to those considering adoption and could easily be provided in the introduction or M&M sections. Alternative, the authors could elect to change the title to reflect the strain engineering focus of the work without reference to platform development.

Authors revisions:

Recommendation 1: We modified the text under “Growth conditions” in the Materials and methods section: “Strains were cultivated in 50 mL of TAP media at 25°C with constant illumination at 80 μmol photons m-2s-1 and agitated at 150 rpm on a rotary shaker.”

Recommendation 2: We elected to change the title to “Engineering the green algae Chlamydomonas incerta for recombinant protein production” to reflect the strain engineering focus of our work.

6. PLOS authors have the option to publish the peer review history of their article (what does this mean?). If published, this will include your full peer review and any attached files.

Do you want your identity to be public for this peer review? For information about this choice, including consent withdrawal, please see our Privacy Policy.

Reviewer #1: No

---

## [Editor Report · Decision Letter 1]

2 Mar 2025

Engineering the green algae Chlamydomonas incerta for recombinant protein production

PONE-D-24-49646R1

Dear Dr. Molino,

We’re pleased to inform you that your manuscript has been judged scientifically suitable for publication and will be formally accepted for publication once it meets all outstanding technical requirements.

Kind regards,

Andrew Webber

Academic Editor

PLOS ONE
---

## [Editor Report · Acceptance letter]

PONE-D-24-49646R1

PLOS ONE

Dear Dr. Molino,

I'm pleased to inform you that your manuscript has been deemed suitable for publication in PLOS ONE. Congratulations! Your manuscript is now being handed over to our production team.

Kind regards,

on behalf of

Dr. Andrew Webber

Academic Editor

PLOS ONE